# Cutaneous Angiosarcoma of the Head and Neck—A Retrospective Analysis of 47 Patients

**DOI:** 10.3390/cancers14153841

**Published:** 2022-08-08

**Authors:** Neeraj Ramakrishnan, Ryan Mokhtari, Gregory W. Charville, Nam Bui, Kristen Ganjoo

**Affiliations:** 1Department of Medicine, Santa Clara Valley Medical Center, 751 S Bascom Ave, San Jose, CA 95128, USA; 2Department of Medicine/Oncology, Stanford Medical Center, 300 Pasteur Drive, Stanford, CA 94305, USA; 3Department of Pathology, Stanford Medical Center, 300 Pasteur Drive, Stanford, CA 94305, USA

**Keywords:** cutaneous angiosarcoma, head and neck, surgery, immunotherapy

## Abstract

**Simple Summary:**

Cutaneous angiosarcoma (CAS) is a rare sarcoma with dismal prognosis. To better characterize this disease and elucidate potential treatments that improve overall survival (OS), we conducted a retrospective study exploring clinical characteristics and treatment outcomes of 47 patients with CAS of the head and neck treated at a tertiary academic center. We found that CAS continues to have a poor prognosis with high rates of recurrence even with current treatment modalities. Surgery was highly effective in improving OS in patients with disease that could be resected with low morbidity. Chemotherapy, radiotherapy (RT), and immunotherapy did not significantly improve OS. Our findings shed light on the current landscape of clinical characteristics and treatment of CAS and could prompt further research exploring new treatment options and role of immunotherapy in the management of this difficult disease.

**Abstract:**

Cutaneous angiosarcoma (CAS) is a rare and aggressive malignant tumor with blood vessel or lymphatic-type endothelial differentiation. It has a poor prognosis with lack of standardized treatment options. This study retrospectively evaluated the clinical characteristics and treatment outcomes of 47 patients with CAS of the head and neck treated at an academic sarcoma center. Patient data were collected from the electronic medical records. 62% of patients were male with the scalp being the most commonly affected area (64%). The majority of patients presented with localized disease (53%). Median overall survival (OS) was 3.4 years with an OS of 36% at 5 years. There was a statistically significant increase in OS for patients who underwent surgery compared to those who did not (5.4 vs. 2.8 years). In contrast, radiotherapy (RT) or chemotherapy did not significantly increase OS. 45% of patients had recurrence of disease during their treatment course with a median time to recurrence of 22.8 months. There was not a significant difference in OS for patients who underwent immunotherapy compared to those who underwent chemotherapy, although only a few patients received immunotherapy. We found that surgery was an effective treatment modality in patients with easily resectable disease, while RT, chemotherapy, and immunotherapy did not significantly improve OS.

## 1. Introduction

Cutaneous angiosarcoma (CAS) is a rare and extremely aggressive malignant neoplasm that originates from vascular endothelial cells of the skin which can arise sporadically, secondary to radiation, or in conjunction with chronic lymphedema (Stewart-Treves Syndrome) [1]. The most common variant of CAS occurs in the head and neck, which is known as Wilson–Jones type. This particular variant typically affects elderly males and represents less than 0.1% of all head and neck cancers [2]. CAS of the head and neck typically demonstrates rapid progression and is associated with the highest rate of lymph node metastases amongst all head and neck soft tissue sarcomas [3]. The most common site of distant metastasis is the lungs, while other less common distal sites of spread include the liver, bone, and soft tissue [2,3]. CAS can be difficult to accurately diagnose given that it may mimic a variety of benign diseases such as hemangiomas or vascular malformations as well as other malignant neoplasms like melanoma [4,5]. CAS has a poor prognosis with a 5 year survival rate ranging from 26–51% [6]. Median overall survival (OS) time has been reported to be 1.8 years with 2 and 5 year survival rates of 47.3% and 26.5% respectively [7]. Factors associated with poor survival rates include: older age, poor performance status, larger tumor size, higher tumor grade, positive margins, deeper tumor depth, and distant metastasis [8,9,10].

Due to the rarity of this disease, challenges associated with its diagnosis, and overall aggressive nature, there remains a lack of optimal standardized treatments. We report our experience regarding the clinical characteristics and treatment outcomes of 47 patients with CAS of the head and neck.

## 2. Materials and Methods

### 2.1. Data Collection

All patients with CAS of the head and neck treated at Stanford between 2000 and 2021 were included in this study. Pathology samples were reviewed by Stanford sarcoma pathologists in real time who confirmed the diagnosis. In addition, the reports were reassessed by Gregory W. Charville for the purpose of this retrospective study, and all diagnoses were reconfirmed. Data, which included patient demographics, tumor characteristics from pathology reports, treatment modalities, and overall outcomes, were collected from Stanford EPIC electronic medical records to build a retrospective database for analysis. Treatment modalities included surgery, radiotherapy (RT), chemotherapy, and immunotherapy. Survival status was assessed through hospital records from Stanford Epic. The date of database lock was 13 December 2021. This study was approved by the Stanford University Institutional Review Board (IRB). Patient consent was not deemed necessary by the IRB due to the retrospective nature of the study as well as de-identification of patient data.

### 2.2. Tumor Classification

Tumor classification was performed using histologic criteria of the WHO Classification of Skin Tumors (fourth edition); CAS was defined as a malignant neoplasm with endothelial differentiation involving the skin [11]. Endothelial differentiation was established morphologically by identification of vasculogenic neoplastic cells.

### 2.3. Statistical Analysis

Statistical analysis was performed using Python 3.0. Survival curves were generated using the Kaplan–Meier Method and the lifelines packages. We compared tumor size at diagnosis and overall survival outcomes in the presence versus absence of the following treatment modalities: surgery, RT, chemotherapy, and immunotherapy. Survival outcomes were also compared between different chemotherapy and immunotherapy regimens. Statistical significance was determined for any *p*-value ≤0.05.

## 3. Results

### 3.1. Patient Characteristics and Treatment Data

A total of 47 patients were eligible and included in this study. Patient characteristics are summarized in Table 1. More patients were male (62%), and the most common primary location was the scalp (64%). The majority of patients presented with localized disease. Median OS for patients with metastatic disease was 2.5 years compared to 3.9 years for those with localized disease (*p* = 0.0795) (Figure 1). About 45% of patients had recurrence of disease during their treatment course. No patients received radiotherapy before their diagnosis of CAS. Overall treatment modalities are summarized in Table 2. Over half of the patients underwent surgical resection as part of their treatment. Half of the patients received chemotherapy (49%) and/or radiation (47%). Median radiation dose in Gy was 63 with a range of 20 to 188. Fifteen percent (*n* = 7) of patients had immunotherapy as part of their treatment regimen. Median length of follow-up was 21.8 months with a range of 0 to 125.4 months and 95% CI [19.4, 36]. About 45% of patients were lost to follow-up from the Stanford Healthcare System.

### 3.2. Evaluation of Outcomes Based on Treatment Modality

The median OS was 3.4 years, ranging from 0.4 to 7.2 years. The OS was 83% and 36% at 2 and 5 years, respectively (Figure 2).

Patients who underwent surgical resection had a statistically significant increase in OS compared to those who did not undergo surgery (5.4 vs. 2.8 years) (*p* = 0.0193) (Figure 3A). There was not a significant difference in OS for patients who had RT (*p* = 0.0696) and chemotherapy (*p* = 0.143) compared to those who did not undergo these treatment modalities. Median OS for those who underwent RT was 3.9 years compared to 2.5 years for those who did not. Median OS for those who underwent chemotherapy was 3.3 years compared to 7.2 years for those who did not (Figure 3B,C). About 45% of patients were found to have positive margins. Median OS of patients who underwent surgery with positive margins was 3.9 years compared to 7.2 years for those with negative margins (0.0559) (Figure 4). Median OS of patients with localized disease who underwent surgery was 5.4 years compared to 3.4 years for patients who underwent RT +/− chemotherapy as well as those who underwent chemotherapy/immunotherapy (Figure 5).

Median time to recurrence was 22.8 months with a range of 1 to 40 months. Median tumor size for patients who underwent surgery versus non-surgical patients was 3.5 and 6 cm, respectively (*p* = 0.02). Patients who underwent RT had larger median tumors (6 cm) compared to non-RT patients (3.5 cm), though this was not statistically significant (*p* = 0.11). Similarly, patients who received chemotherapy had larger median tumors (6 cm) compared to non-chemotherapy patients (3.5 cm), which was not statistically significant (*p* = 0.08).

### 3.3. Comparisons of Outcomes between Chemotherapy and Immunotherapy

The most commonly used chemotherapy agents included paclitaxel (*n* = 26), gemcitabine (*n* = 8), and pazopanib (*n* = 7). Median OS for paclitaxel, gemcitabine, and pazopanib were 5.1 months, 3.2 months, and 3.3 months respectively, though this was not statistically significant (Figure 6).

Notably, there was not a significant difference in OS for patients who underwent immunotherapy compared to those who underwent chemotherapy/targeted therapy (*p* = 0.5921). Median OS for those who underwent immunotherapy was 3 months compared to 4 months for those who underwent chemotherapy (Figure 7).

## 4. Discussion

This study reports the overall clinical characteristics and outcomes of patients with CAS treated at a large academic center. In our study the majority of patients were male, older than 70 years of age, and Caucasian. Additionally, 53% of patients presented with local disease which is consistent with the current literature [7,12]. Angiosarcoma can present cutaneously (cAS) or non-cutaneously (NC-AS). Each form of angiosarcoma presents distinctly. In a study comparing treatment outcomes for patients with CAS versus non-CAS, median age at diagnosis was significantly lower (58 years) with a more equal distribution between genders for patients with non-CAS. While patients with CAS tended to present locally, half of patients with non-CAS presented with metastatic disease at diagnosis. There were no significant differences in OS after both groups underwent similar treatment modalities of surgery, radiation, and paclitaxel as the most common first line chemotherapy [13]. A distinct feature of CAS is the potential presence of UV mutational signatures that can influence response to immunotherapy [14]. A retrospective analysis of solid tumors analyzed by comprehensive genomic profiling between December 2013 and June 2021 found 82 angiosarcomas to have UV mutational signatures [15]. A genomic analysis of 48 angiosarcoma samples also found that face and scalp angiosarcomas were more specifically impacted by mutations attributable to UV radiation. This particular UV mutational signature and overall mutational burden are important potential biomarkers for response to immunotherapy treatment [16].

We report survival outcomes similar to retrospective studies conducted at tertiary academic centers [12,17]. On the other hand, OS significantly improved compared to those reported in studies analyzing data from Surveillance, Epidemiology, and End Results program of the National Cancer Institute [7,18]. This could highlight the importance of prompt referral of patients with rare disease to tertiary academic centers to maximize treatment options and efficacy. Notably, OS has been found to be significantly worse for scalp versus facial CAS in the current literature [19]. Other forms of CAS include those that present on the trunk/extremities, radiation induced angiosarcoma of the breast, and Stewart–Treves Syndrome. A retrospective study analyzing clinical outcomes of angiosarcomas in various locations found that median OS (21.7 months) was longest in patients with CAS located on the trunk [20].

Survival outcomes of radiation induced angiosarcoma of the breast are comparable to those of head/neck CAS [21,22]. On the other hand, survival outcomes of Stewart–Treves Syndrome are worse compared to those of head/neck CAS with a median survival of 2.5 years after diagnosis and dismal 5-year survival rates between 8.5% and 13.6% [23].

First line treatment of CAS is currently surgical resection with postoperative radiotherapy. It is important to note that surgery can be quite morbid for patients with extensive disease and high rates of local recurrence. As a result, non-invasive treatment with modalities such as radiation and/or chemotherapy can be considered [24]. We found that patients who underwent surgical resection had a significant increase in OS (5.4 years) compared to those who did not undergo surgery (2.8 years). This is consistent with the current literature illustrating surgery as a definitive and effective treatment modality [25,26]. It is important to acknowledge that patients in our study who underwent surgery had smaller tumors (3.5 cm) compared to those who did not undergo surgery (6 cm). Given that patients who underwent surgery had potentially less advanced disease, the benefit of surgical intervention could be skewed as it is done in selected patients with small tumors. Thus, there is a selection bias that could explain the better outcomes of surgery in comparison to other treatment modalities. We attempted to minimize this selection bias by comparing OS between treatment modalities only in patients with localized disease. In this particular subset of patients with localized disease, surgery still showed improved OS compared to other treatment modalities of RT +/− chemotherapy and chemotherapy/immunotherapy. Data regarding the utility of radiotherapy has been mixed. Interestingly, our study did not find a significant difference in OS for patients who underwent RT compared to those who did not. This could be confounded by the fact that patients who underwent RT had larger tumors (6.5 cm) and hence potentially more advanced disease, compared to those who did not undergo RT (3.5 cm). While a retrospective study showed that RT did not significantly improve OS for patients with cAS, many previous studies have found that RT significantly improved both local control and OS [17,26].

Paclitaxel is currently first line systemic treatment for unresectable, recurrent, or metastatic CAS, with the ANGIOTAX prospective clinical trial initially confirming its beneficial role [27]. Several studies have even demonstrated significantly higher OS and progression free survival (PFS) with paclitaxel and radiotherapy followed by maintenance taxane chemotherapy compared to conventional surgery and radiotherapy. Moreover, this particular treatment modality could especially be invaluable in cases of large tumors that cannot be excised easily surgically [28,29].

Other systemic treatment options include anti-VEGF drugs such as bevacizumab and pazopanib, gemcitabine, anthracyclines, eribulin, regorafenib, and propranolol [6].

Bevacizumab, in particular, has been shown to be an effective treatment modality by demonstrating a PFS of 6.5 months in a phase II clinical study [30]. Another randomized phase II trial displayed the same PFS and similar OS between CAS patients receiving bevacizumab and paclitaxel and those only undergoing treatment with paclitaxel. However, overall toxicity and serious adverse events were significantly higher in the treatment arm undergoing combination treatment compared to those receiving monotherapy with paclitaxel [31]. Regorafenib, a small molecule multikinase inhibitor, has also emerged as an effective treatment modality particularly in cases of refractory metastatic and unresectable CAS [32]. Interestingly, propranolol displayed growth attenuation of CAS through a reduction in the proliferative index of the tumor in a case report. Subsequent combination of propranolol with paclitaxel and RT ultimately led to extensive tumor regression and no detectable metastasis in this case as well [33]. Our study reported a wide array of agents including cytotoxic chemotherapy combinations, anti-VEGF drugs, and immunotherapy. Gemcitabine, paclitaxel, and pazopanib were the most commonly used agents with no significant difference between them in terms of OS. Moreover, our study did not illustrate any significant difference between patients who underwent chemotherapy versus those who did not. Similar to outcomes regarding RT, this could be confounded by the fact that patients who underwent chemotherapy had more advanced disease (median tumor size: 6 cm) compared to those who did not (median tumor size: 3.5 cm). Our findings are in agreement with the current literature, which indicates that adjuvant chemotherapy after surgery does not improve OS [17,34,35].

CAS is molecularly heterogeneous. Primary CAS (pCAS) contains mutations in TP53 as well as several mutations of the MAP kinase pathway such as KRAS, HRAS, NRAS, BRAF, MAPK1, and NF1. Contrastingly, secondary CAS (sCAS) contains mutations in genes such as KIT, FLT4, RET, and CTLA4. There is a strong overexpression of Myc exclusively in sCAS, while there is Myc amplification/overexpression in some cases of pCAS. Moreover, TP53 loss of function and MYC amplification/overexpression occur in an almost mutually exclusive manner in pCAS and sCAS respectively. A study that investigated multiomic analysis of angiosarcoma subdivided pCAS into two clusters with low (cluster 1) or high (cluster 3) tumor inflammation signatures (TIS). About 50% of cases in both cluster 1 and cluster 3 displayed UV mutational signals and high tumor mutational burden (TMB). Cluster 3 patients with high TMB and TIS scores represent the population with highest probability for efficacious response rates from immune checkpoint inhibitor treatments [36]. Moreover, immunotherapy has shown promise as a treatment modality in several studies. For instance, a case series demonstrated that 5 out of 7 patients with angiosarcoma achieved either partial or complete response with immune checkpoint inhibitor therapy [37]. Another retrospective study also showed efficacy of immune checkpoint blockade against CAS of the head and neck, particularly those with a dominant mutational signal associated with ultraviolet light [38]. A study exploring combination use of ipilimumab and nivolumab for metastatic or unresectable angiosarcoma demonstrated an overall response rate of 25% with response from 3 out of 5 patients with cutaneous tumors of the scalp or face [39]. In our study 7 out of 47 patients were treated with immunotherapy, namely nivolumab and ipilimumab. Moreover, there was no significant difference in OS for those who underwent immunotherapy compared to those who underwent chemotherapy. This result may be due to a combination of small sample size and advanced disease by the time of initiation of immunotherapy.

Overall, our study supports surgical resection as the preferred modality of treatment for CAS that can be resected with low morbidity. Given that our study was retrospective in nature, surgery was not commonly offered to patients with extensive disease and more heavily considered for those with resectable disease. Interestingly, our study did not display any significant difference in OS for patients who underwent radiation, chemotherapy, and even immunotherapy compared to those who did not receive these treatment modalities. Small sample size and advanced disease in those who received immunotherapy could be reasons why this treatment modality did not significantly improve OS compared to other studies. Nevertheless, increasing the range of patients who receive immunotherapy and also tailoring it to particular demographics such as those who possess a dominant mutational signal associated with ultraviolet light could potentially make a significant impact with regard to improving outcomes of CAS. Current ongoing clinical trials exploring therapies outside of immunotherapy include intralesional injection of talimogene laherparepvec, regorafenib, and combination TRC105 [40,41,42]. Moreover, further research is needed in regard to assessing larger scale interventions of CAS involving immunotherapy.

Our study has several limitations. First, it is subject to the inherent biases of a retrospective study design such as selection bias, given that patients with more complex or advanced disease were more likely to undergo heterogeneous treatment. Additionally, genomic profiling data were not available for our subset of patients. Certain information regarding treatment outcomes was also unavailable given that certain patients pursued medical care at outside institutions subsequently after referrals to Stanford University. Small sample size also poses a limitation.

## 5. Conclusions

Our findings indicate that CAS has poor outcomes with high rates of recurrence. Surgery is a treatment modality that significantly improves OS in patients with disease that can easily be resected with low morbidity. RT, chemotherapy, and immunotherapy did not significantly improve OS for patients in our study. Despite this, immunotherapy still represents a treatment modality that could improve outcomes in CAS based on other studies. CAS patients with UV mutational signatures can potentially benefit from immunotherapy. Furthermore, new treatment options and additional studies exploring the utility of immunotherapy in the management of CAS are needed.

## Figures and Tables

**Figure 1 cancers-14-03841-f001:**
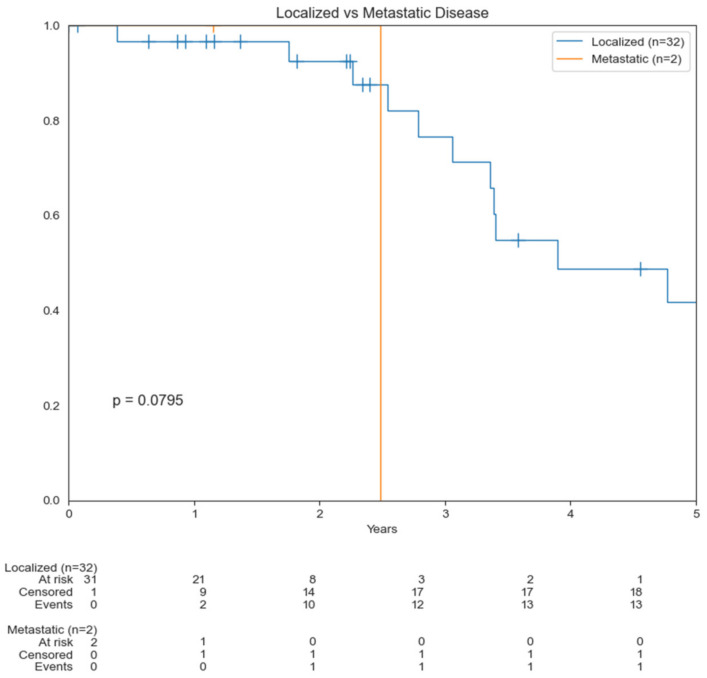
Kaplan–Meier curves comparing overall survival (OS) based on localized disease versus metastasis.

**Figure 2 cancers-14-03841-f002:**
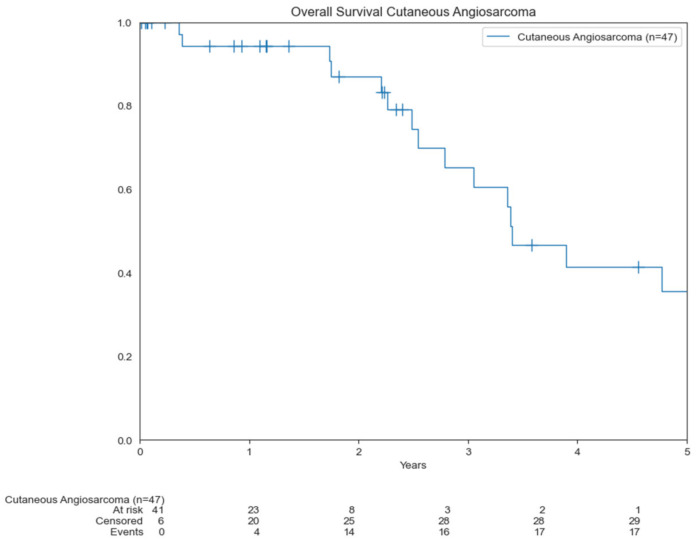
Kaplan–Meier curve showing OS for patients.

**Figure 3 cancers-14-03841-f003:**
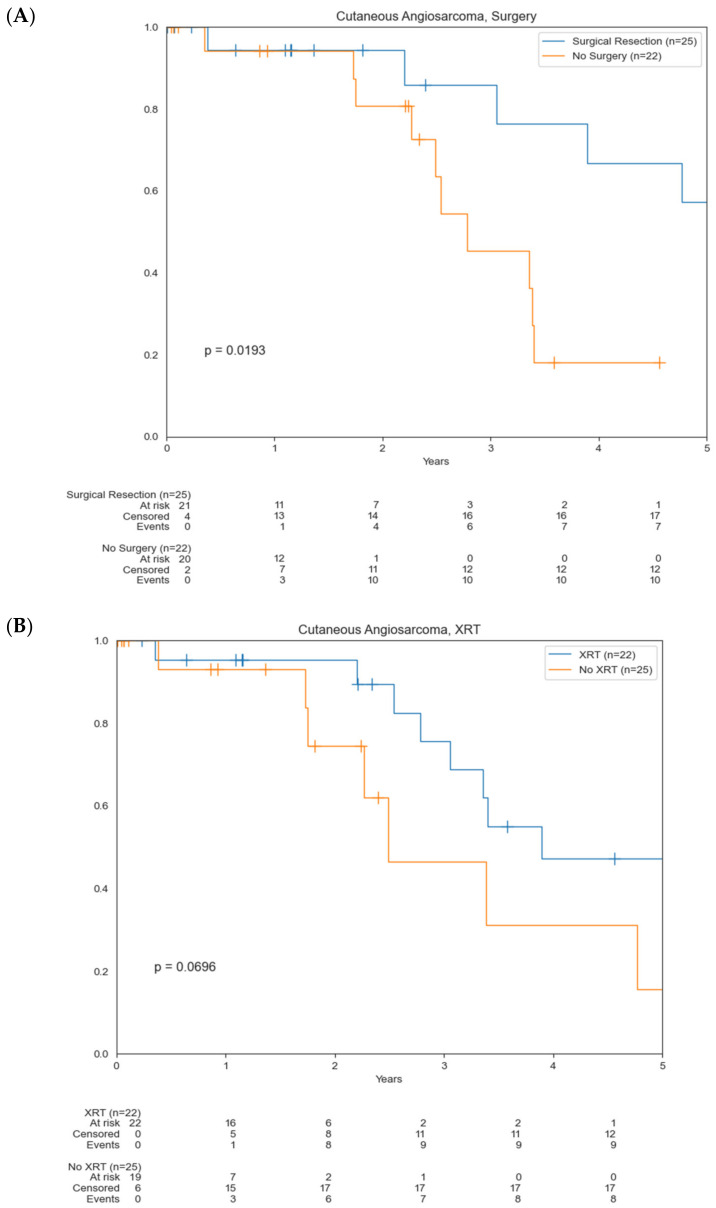
Kaplan–Meier curves comparing OS based on treatment modality. (**A**) Surgical resection versus no surgery; (**B**) radiotherapy (RT) versus no RT; (**C**) chemotherapy versus no chemotherapy.

**Figure 4 cancers-14-03841-f004:**
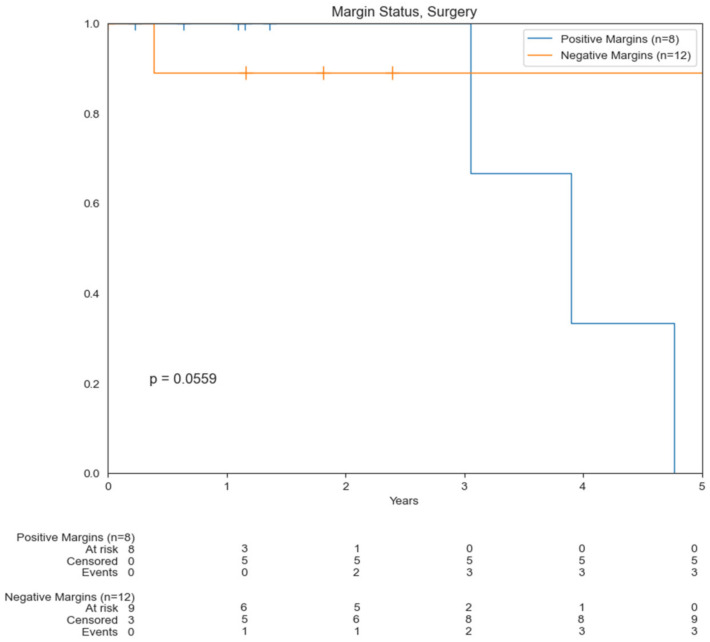
Kaplan–Meier curves comparing OS based on margin status.

**Figure 5 cancers-14-03841-f005:**
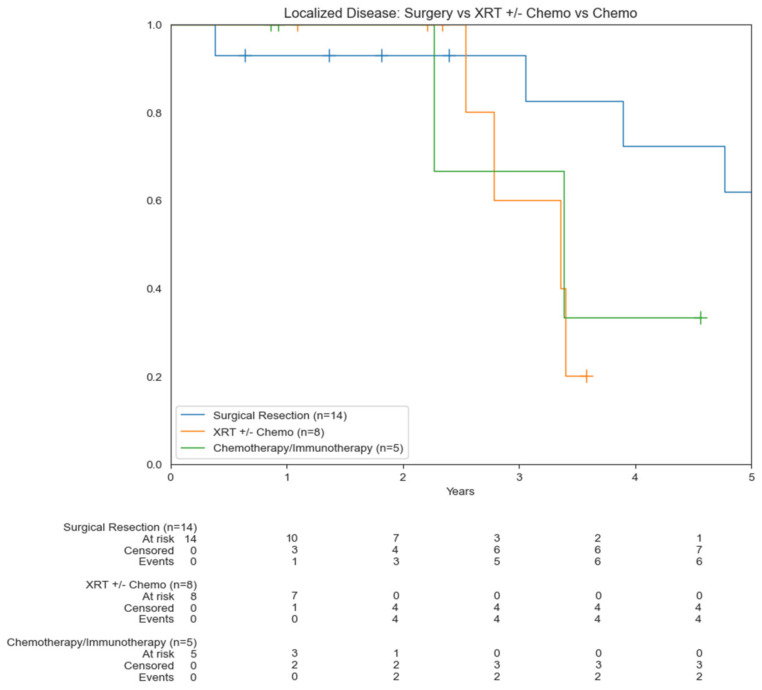
Kaplan–Meier curves comparing OS based on treatment modality in localized disease.

**Figure 6 cancers-14-03841-f006:**
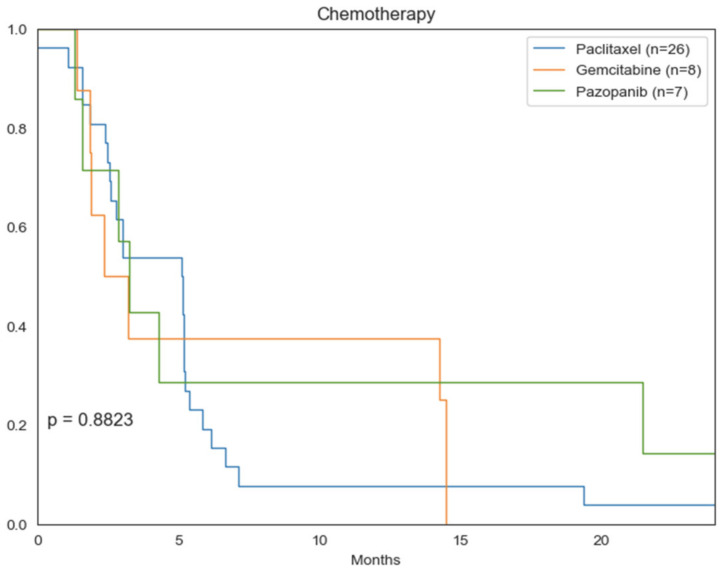
Kaplan–Meier curves comparing OS between chemotherapy regimens.

**Figure 7 cancers-14-03841-f007:**
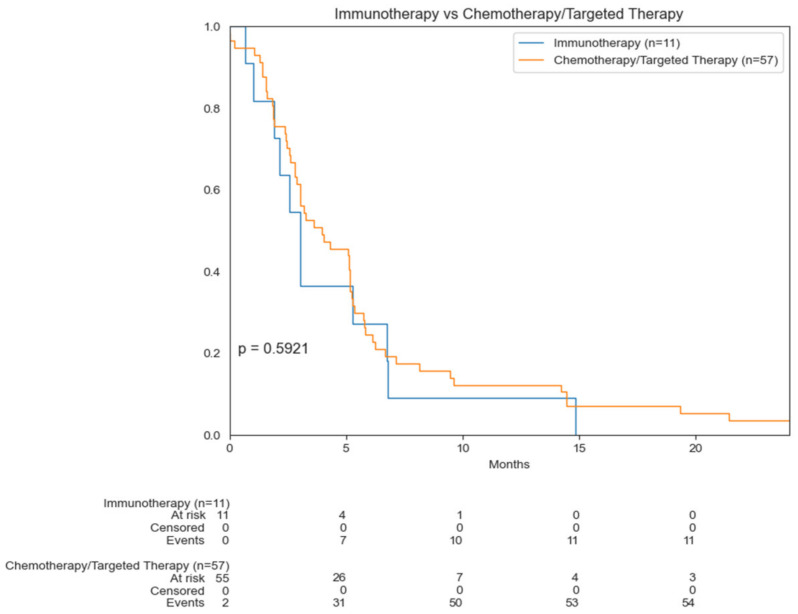
Kaplan–Meier curves comparing OS of immunotherapy versus chemotherapy/targeted treatment.

**Table 1 cancers-14-03841-t001:** Summary of patient characteristics.

Characteristics	Frequency (*n*)	Percentage (%)
**Total (overall)**	47	100%
**Gender**		
Male	29	62%
Female	18	38%
**Age Group (Years)**		
<60	3	6%
60−79	9	19%
70−79	19	40%
80−89	16	34%
**Race**		
Caucasian	20	43%
Asian	10	21%
African American	1	2%
Hispanic	2	14%
Other	14	30%
**Location**		
Scalp	30	
Cheek	7	
Forehead	7	
Post-Auricular/Ear	4	
Orbital/Eyelid	3	
Nose	2	
Neck	1	
**Local versus Metastatic (Entire Treatment Course)**		
Local	25	53%
Metastatic	11	23%
Unknown	11	23%
**Stage (At Diagnosis)**		
1	23	49%
2	6	13%
3	7	15%
4	2	4%
Unknown	9	19%

**Table 2 cancers-14-03841-t002:** Summary of treatment data.

Treatment	*n* (%)	Mediation Duration (Months)
**Treatment Type**		
Surgery	25 (53)	
Radiation	22 (47)	
Chemotherapy	23 (49)	
Immunotherapy	7 (15)	
**Regimen (Adjuvant)**		
Paclitaxel	20 (43)	
Pazopanib	7 (15)	
Gemcitabine	4 (9)	
Doxorubicin	4 (9)	
Nivolumab	3 (6)	
Interferon Alpha	1 (2)	
Abraxane	1 (2)	
Gemcitabine + Docetaxel	1 (2)	
Gemcitabine + Paclitaxel	1 (2)	
Carboplatin + Paclitaxel	1 (2)	
Adriamycin + Olaratumab	1 (2)	
Doxorubicin + Eribulin	1 (2)	
**Regimen (Metastatic)**		
Gemcitabine	4 (9)	2.5
Paclitaxel	3 (6)	3.93
Pazopanib	2 (4)	2.86
Nivolumab	2 (4)	1.87
Ipilimumab	1 (2)	2.14
Doxorubicin	1 (2)	1.41
Gemcitabine + Docetaxel	2 (4)	3.27
Nivolumab + Doxorubicin	1 (2)	1.02
Gemcitabine + Paclitaxel	1 (2)	6.21
Nivolumab + Ipilumumab	1 (2)	14.66
Nivolumab + Pazopanib	1 (2)	5.26
Adriamycin + Olaratumab	1 (2)	0.23
Nivolumab + Gemcitabine + Paclitaxel	1 (2)	2.56

## Data Availability

The data presented in this study are available on request from the corresponding author. The data are not publicly available due to the fact that it was manually collected from the electronic medical records and de-identified.

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
