# Peer review of "Cutaneous Angiosarcoma of the Head and Neck—A Retrospective Analysis of 47 Patients"

_cancers, 2022, doi:10.3390/cancers14153841_

Round 1
Reviewer 1 Report
The manuscript needs more depth. Authors insist that radiotherapy or chemotherapy is not effective for CAS, but surgery is effective. However, 45% of patients were found to have positive margin. In conclusion, authors insist the possibility that surgery was performed for patients with early stage. Therefore, I think that authors should indicate the content of the treatment and outcome according to the stage. I think that surgery with postoperative radiotherapy is gold standard for patients with early stage, and that the successful treatment for patients with advanced stage does not exist.
Author Response
Point 1: The manuscript needs more depth. Authors insist that radiotherapy or chemotherapy is not effective for CAS, but surgery is effective. However, 45% of patients were found to have positive margin. In conclusion, authors insist the possibility that surgery was performed for patients with early stage. Therefore, I think that authors should indicate the content of the treatment and outcome according to the stage. I think that surgery with postoperative radiotherapy is gold standard for patients with early stage, and that the successful treatment for patients with advanced stage does not exist.
Response 1: We acknowledge and agree with your point that surgery with postoperative radiotherapy is optimal treatment for patients with early stage disease and that successful treatment for patients with advanced stage currently does not exist. To further illustrate this and address treatment and outcome according to stage, we added a Kaplan Meier Curve comparing OS of patients with localized disease versus metastatic disease. We also added a figure comparing OS based on margin status to further characterize the importance of margin status in regard to surgical treatment outcomes.
Reviewer 2 Report
Cutaneous Angiosarcoma (CAS) is a rare and aggressive malignant tumor with blood vessel or lymphatic-type endothelial differentiation. It has a poor prognosis with lack of standardized treatment options.
This study retrospectively evaluated the clinical characteristics and treatment outcomes of 47 patients with CAS of the head and neck treated at an academic sarcoma center. It adds little to the clinical knowledge on that rare tumor. 62% of patients were male with the scalp being the most commonly affected area (64%), as previously reported. The reader should know whether some patients received prior radiotherapy (post-radiotherapy angiosarcoma??)
However, the paper is well written, and the therapeutic part deserves publication provided improvements to the manuscript are made.
The majority of patients presented with localized disease (53%). Median overall survival (OS) was 3.4 years with an OS of 36% at 5 years. There was a statistically significant increase in OS for patients who underwent surgery compared to those who did not (5.4 vs 2.8 years). In contrast, radiotherapy (RT) or chemotherapy did not significantly increase OS. 45% of patients had a recurrence of disease during their treatment course with a median 30 time to recurrence of 22.8 months. There was no significant difference in OS for patients who underwent immunotherapy compared to those who underwent chemotherapy, although only a few 32 patients received immunotherapy.
The reader needs the following information that I was unable to find: Date of database lock. Length of follow-up (median, 95%CI and range), and the number of patients lost to follow-up. How was survival status assessed? From hospital records, or from death certificates?
For all Kaplan Meier figures, we need to see the numbers at risk below the abscise +++
Did the presence of positive margins after surgery modify OS? Were the presence of positive margins the reasons for postoperative radiotherapy? In fact, the readers need more pieces of information on radiotherapy (indication, doses, fractions etc) as outcomes after post-operative radiotherapy were borderline significative
The discussion section is a little bit outdated for systemic treatment
• Bevacizumab, a VEGFR inhibitor, was reported to be an effective with a PFS of 6.5 months.
• Combinations of bevacizumab and paclitaxel: no benefit, but toxicity +++.
• Regorafenib, a dual inhibitor of VEGFR2-TIE2 tyrosine kinase was also used
• Propanolol: one case record alone, and the addition of propranolol to the chemotherapy regimen has shown promising response in several case reports.
Minor concerns:
Table 2 can be simplified by regrouping row with 1 patient.
Fig 3 can be deleted, with results given in the text
Author Response
Point 1: The reader should know whether some patients received prior radiotherapy (post-radiotherapy angiosarcoma??)
Response 1: We have now indicated that none of our patients received radiotherapy prior to their diagnosis of CAS.
Point 2: The reader needs the following information that I was unable to find: Date of database lock. Length of follow-up (median, 95%CI and range), and the number of patients lost to follow-up. How was survival status assessed? From hospital records, or from death certificates?
Response 2: We added the date of database lock, length of follow-up (median, 95% CI, and range), as well as number of patients lost to follow-up. Additionally, we have now indicated that survival status was assessed from hospital records through Stanford EPIC.
Point 3: For all Kaplan Meier figures, we need to see the numbers at risk below the abscise +++.
Response 3: We have accordingly edited the Kaplan Meier Figures with the exception of figure 5. It was technically difficult to include the numbers at risk below the abscise for figure 5.
Point 4: Did the presence of positive margins after surgery modify OS? Were the presence of positive margins the reasons for postoperative radiotherapy? In fact, the readers need more pieces of information on radiotherapy (indication, doses, fractions etc) as outcomes after post-operative radiotherapy were borderline significative
Response 4: We have created a figure that compares OS of patients with positive margins versus negative margins after surgery. There was no particular pattern in regard to indication for radiotherapy that we could find in our patient population, so we did not include this. However, we did add median and range in regard to RT dose.
Point 5:
The discussion section is a little bit outdated for systemic treatment
- Bevacizumab, a VEGFR inhibitor, was reported to be an effective with a PFS of 6.5 months.
- Combinations of bevacizumab and paclitaxel: no benefit, but toxicity +++.
- Regorafenib, a dual inhibitor of VEGFR2-TIE2 tyrosine kinase was also used
- Propanolol: one case record alone, and the addition of propranolol to the chemotherapy regimen has shown promising response in several case reports.
Response 5: We added sources that addressed these above points in our discussion section.
Point 6: Table 2 can be simplified by regrouping row with 1 patient.
Response 6: We felt that regrouping table 2 in this fashion may actually make it more complex, so we kept the table layout as is.
Point 7: Fig 3 can be deleted, with results given in the text
Response 7: We have deleted Fig 3.
Reviewer 3 Report
This study presents data from 47 patients treated for cutaneous angiosarcoma, a rare sarcoma. This study is very well written, analysed and presented with a very balanced discussion and awareness of limitations. It was a pleasure to read and review this quality report. I have two comments, the first of which is an essential edit.
1) The authors state ethical approval but no statement is provided about participant consent. Please state whether written informed consent was obtained for this study. Understanding that this is a retrospective study, were surviving patients consented?
2) Given the report shows surgical success and discusses the consideration that tumours chosen for surgical intervention are those that are smaller with greater potential for resection, can the authors please add stage at diagnosis compared with O/S for surgery versus non-surgical intervention? It would seem given the results presented that this graph may highlight the need for early diagnosis and (as noted in Discussion) referral to academic / medical centres of expertise.
Finally, congratulations to the Stanford centres on their comparatively high O/S in this disease compared to many global centres.
Author Response
Point 1: The authors state ethical approval but no statement is provided about participant consent. Please state whether written informed consent was obtained for this study. Understanding that this is a retrospective study, were surviving patients consented?
Response 1: Patient consent was not deemed necessary by the IRB due to the retrospective nature of the study as well as de-identification of patient data. We have now included this in the methods section.
Point 2: Given the report shows surgical success and discusses the consideration that tumours chosen for surgical intervention are those that are smaller with greater potential for resection, can the authors please add stage at diagnosis compared with O/S for surgery versus non-surgical intervention? It would seem given the results presented that this graph may highlight the need for early diagnosis and (as noted in Discussion) referral to academic / medical centres of expertise.
Response 2:
We added a Kaplan Meier Curve that compares OS for patients with localized disease to those with metastatic disease to further characterize influence of stage on outcome. To more specifically characterize surgical outcomes, we also added a Kaplan Meier Curve that compares OS for patients based on margin status.
Reviewer 4 Report
This is a nice study with clinical interest. This retrospective study indicates that surgery is an effective treatment modality in patients with easily resectable disease. Several studies from Japan suggest that chemotherapy (paclitaxel or docetaxel) + radiation (RT) followed by maintenance taxane chemotherapy could be a better therapeutic option than other interventions, especially in the patients with large tumor sizes. Would you please add some references to discussion section?
1) Seo, Kitamura et al. Efficacy of a combination of paclitaxel and radiation therapy against cutaneous angiosarcoma: a singly institution retrospective study of 21 cases. J Dermatol 2022; 44: 383-386.
2) Fujisawa, Yoshino et al. Chemoradiotherapy with taxane is superior to conventional surgery and radiotherapy in the management of cutaneous angiosarcoma: a multicentre, retrospective study. Br J Dermatol 2014: 171: 1493-1500.
Author Response
Point 1: This is a nice study with clinical interest. This retrospective study indicates that surgery is an effective treatment modality in patients with easily resectable disease. Several studies from Japan suggest that chemotherapy (paclitaxel or docetaxel) + radiation (RT) followed by maintenance taxane chemotherapy could be a better therapeutic option than other interventions, especially in the patients with large tumor sizes. Would you please add some references to discussion section?
1) Seo, Kitamura et al. Efficacy of a combination of paclitaxel and radiation therapy against cutaneous angiosarcoma: a singly institution retrospective study of 21 cases. J Dermatol 2022; 44: 383-386.
2) Fujisawa, Yoshino et al. Chemoradiotherapy with taxane is superior to conventional surgery and radiotherapy in the management of cutaneous angiosarcoma: a multicentre, retrospective study. Br J Dermatol 2014: 171: 1493-1500.
Response 1: We have accordingly included these references in our discussion section.
Round 2
Reviewer 2 Report
The issues I have adressed have been correctly adressed by the authors. The manuscript has improved significantly
Author Response
Thank you kindly for your feedback, it was greatly appreciated.